# The influence of environment on bacterial co-abundance in the gut microbiomes of healthy human individuals
Christophe Boetto [1] ✉, Violeta Basten Romero[1], Léo Henches[1], Arthur Frouin[1], Antoine Auvergne[1],
Etienne Patin [2], Marius Bredon[3], Milieu Intérieur Consortium*, Sean P. Kennedy [1], Darragh Duffy [4],
Lluis Quintana-Murci[2,5], Harry Sokol[3,6,7] & Hugues Aschard [1,8] ✉

The gut microbiome is a complex ecosystem characterized not only by its marginal taxonomic composition but also by its emergent properties. Bacteria develop local interactions to form coherent functional communities, whose effects on health and diseases cannot be predicted from the behavior of individual members. Understanding the factors underlying variability in these communities may therefore provide critical insights on the biological links between the gut microbiome and human phenotypes. Here, we examined the effect of a range of host factors, including demographics, medical history, and dietary habits, on these communities in 938 healthy individuals using MANOCCA, a covariance-based approach developed to address existing limitations. Increased age and smoking were associated with a significant overall decrease in co-abundance, and conversely a higher body mass index was associated with increased co-abundance. At the taxon level, a core of 200 genera were systematically impacted in their co-abundance with other taxa, suggesting a central role in structuring the network. Finally, we demonstrate that our approach offers a powerful framework for prediction purposes, with taxa co-abundance being able to predict the age of participants with an accuracy three-fold higher than a model based on abundance only.

There is an increasing corpus of research describing features associated with the gut microbiome composition in both healthy individuals and disease cases[1–4]. Prominent features include age[5,6], sex[7–9], long term diet[10–12], and host genetics[13–16], but many other factors have been found associated[17,18]. Yet, some aspects of the host-microbiome relationship remain difficult to characterize[19,20]. In particular, handling the high-dimensional nature of microbiome data presents methodological challenges. As a result, most existing studies have focused on univariate approaches, evaluating the relationship between single taxa and host features one by one. Moving toward more comprehensive approaches that can capture the dynamics, connectivity, and high-order characteristics of the microbiome is an active area of research. Beyond established descriptive multivariate metrics—such as alpha and beta diversity indexes, which measure intra- and inter-sample distances between microbiome samples[13,21]—various multivariate methods

have been proposed to jointly test associations between host factors and the abundance of multiple taxa[22–24]. One emerging research topic is the study of the co-abundance of taxa in metagenomes[25–27], and in gut microbiome in particular[28–32].

The gut microbiome is a complex ecosystem whose constituents form sub-communities through interactions between individual taxa. Those sub-communities, sometimes referred to as guilds[29], or cliques, display co-abundances because they exploit the same class of resources or work together as a coherent functional group[25,30]. Studying taxa co-abundance—and more generally, the global connectivity within the gut microbiome—can help identify emergent properties that would be missed by univariate methods. Previous studies already illustrated differences in co-abundance networks across inflammatory bowel disease status and body mass index[28], and geographically diverse populations[31]. However, there is currently no

[1]Institut Pasteur, Université Paris Cité, Department of Computational Biology, F-75015 Paris, France. [2]Human Evolutionary Genetics Unit, Institut Pasteur, Université Paris Cité, CNRS UMR2000, 75015 Paris, France. [3]Sorbonne Université, INSERM UMRS-938, Centre de Recherche Saint-Antoine, CRSA, AP-HP, Paris, 75012, France. [4]Translational Immunology Unit, Institut Pasteur, Université de Paris Cité, 75015 Paris, France. [5]Chair of Human Genomics and Evolution, Collège de France, 75005 Paris, France. [6]Gut, Liver & Microbiome Research (GLIMMER) FHU, Paris, France. [7]Université Paris-Saclay, INRAe, AgroParisTech, Micalis Institute, Jouy-en-Josas, 78350, France. [8]Department of Epidemiology, Harvard TH Chan School of Public Health, Boston, MA, 02115, USA. *A list of authors and their affiliations appears at the end of the paper. ✉e-mail: boetto.christophe@gmail.com; hugues.aschard@pasteur.fr

gold standard method to screen for factors associated with changes in the co-abundance of bacteria across individuals in a population. Existing approaches are typically based on the inference of taxa networks across conditions using a threshold-based design to define significant co-abundances when the phenomenon is likely continuous. Furthermore, previous works showed that the inferred networks can vary substantially across approaches[33]. More problematic, by construction those methods are restricted to categorical predictors, they do not allow for covariate adjustment, and the significance of the observed association is typically derived through permutations.

The recently developed MANOCCA[34] test offers a formal statistical framework to test the effect of both categorical and continuous predictors on the covariance matrix of a multivariate outcome—a metric directly proportional to the co-abundance—thus addressing all these methodological limitations. Briefly, the MANOCCA models pairwise taxa covariances at the individual level (i.e. each individual has its own taxa covariance matrix) and use them as dependent variables in a standard multivariate regression to evaluate their association with predictors of interest. Existing tools[35] assume the observed covariances for a given discrete condition (e.g. male or female) are drawn from a single underlying distribution and aim to infer that distribution. In comparison, the MANOCCA treats the covariances as random variables that varies across environmental factors, and aims at estimating the effect of those factors. Note that by construction, the test is independent of the baseline covariance, and does not aim at identifying strongly co-abundant taxa whose interactions is constant across environmental conditions. Here, we applied MANOCCA to characterize the host factors associated with variability in the gut microbiome co-abundance network of healthy participants. We further performed an in-depth examination of the effect of each associated factor on the taxa interaction network, highlighting the key taxa impacted and how these factors shape the microbiome co-abundance. We conclude by illustrating how the proposed framework can be used to develop predictive models of host factors based on taxa co-abundances.

## Results

### Identifying host factors associated with changes in the taxa co-abundance structure

We evaluated variability in the gut microbiome co-abundance structure of healthy individuals conditional on host factors using samples from the Milieu Intérieur (MI) consortium[36]. MI is a population-based cohort including 1,000 healthy participants recruited in France, and split equally by sex and stratified across five-decades of life (from 20 to 60 years old). Gut microbiota composition was obtained from shotgun metagenomics sequencing, and taxonomic levels were derived by summing the normalized abundances (Fig. S1). We used MANOCCA to screen for association between 80 host features collected at baseline, including demographics, dietary habits, medical history and biomarkers measurements (Supplementary Data 1), and taxa co-abundance at the species, genus, and family levels using 938 MI participants with complete data (Fig. 1a-b). We focused on the most common taxa[37] and where MANOCCA maintained calibration despite strong data sparsity (Fig. S1c-d) for a total of 675 species (40% occurrence in samples), 718 genus (5% occurrence in samples), and 151 families (5% occurrence in samples), and after quality control filtering (Supplementary Data 2). For each feature and each taxonomic level, a single MANOCCA test was performed, evaluating if the overall co-abundance (for example the 675 ×675 covariance matrix at the species level) varies with that feature. Except when used as predictors, all analyses were adjusted for age, sex, and body mass index (BMI).

MANOCCA identified associations at a stringent Bonferroni correction threshold ($P < 6.25 \times 10^{-6}$) at all three taxonomic levels with age ($P_{species} = 2.0 \times 10^{-55}$, $P_{genus} = 3.5 \times 10^{-56}$, $P_{family} = 9.2 \times 10^{-37}$), sex ($P_{species} = 2.2 \times 10^{-17}$, $P_{genus} = 6.3 \times 10^{-22}$, $P_{family} = 3.1 \times 10^{-18}$), smoking ($P_{species} = 2.8 \times 10^{-14}$, $P_{genus} = 1.6 \times 10^{-20}$, $P_{family} = 5.6 \times 10^{-13}$), and at the genus level for BMI ($P_{species} = 1.6 \times 10^{-5}$, $P_{genus} = 5.9 \times 10^{-6}$, $P_{family} = 5.3 \times 10^{-5}$) (Fig. 1c-d and Fig. S2). Several features displayed suggestive significant association ($P < 6.25 \times 10^{-4}$), including appendicectomy ($P_{genus} = 6.3 \times 10^{-6}$,

$P_{family} = 4.2 \times 10^{-5}$), cholesterol ($P_{species} = 8.6 \times 10^{-6}$) and dairy products ($P_{genus} = 8.1 \times 10^{-6}$), and at a slightly lower level, raw fruits, Nutrinet factor 1, income, dried pulses, cured cooked meats, bread, steady job, traditional nutrient profile, typhoid vaccination and level of urea. Besides significant and suggestive significant signals, we observed an enrichment for association with taxa co-abundance for nutrition-related features suggesting a modest but systematic impact of diet on the taxa interaction network. For example 23 out of the 30 variables of the diet panel, and 5 of the 7 factors of the Nutrinet panel displaying association below the nominal significance level of 0.05 at the genus level (Supplementary Data 3). We also conducted sensitivity analyses, assessing the variability in the results when applying MANOCCA to random subsets of available taxa. Overall, the larger the set of taxa included, the stronger the association signal (Fig. S2e-h), highlighting a global effect of the four factors on the microbiome co-abundance network.

We applied two alternative multivariate approaches for comparison purposes: a standard MANOVA, testing for association between each host factor and the joint abundance of taxa, and alpha diversity using the Shannon and Simpson indices (Fig. S2i-k, Supplementary Data 3). Some factors were significant after correction for multiple testing but at a much lower significance level. The MANOVA identified an association with age at the species and family level ($P_{species} = 5.2 \times 10^{-6}$, $P_{family} = 1.0 \times 10^{-29}$) and with sex at the family level ($P = 5.2 \times 10^{-10}$). Both Simpson and Shannon indexes identified a signal with age at the family level ($P_{Simpson} = 3.4 \times 10^{-4}$, $P_{Shannon} = 3.9 \times 10^{-5}$) and the Shannon index also identified age at the genus level ($P_{Shannon} = 1.6 \times 10^{-5}$). Altogether, the signals observed exclusively with MANOCCA for smoking and BMI, and the stronger association of age and sex as compared to mean-based and diversity index approaches, points towards a substantially larger information content of the co-abundances of taxa over these existing metrics to describe the relationship between the gut microbiome of healthy individuals and these host variables.

### Contribution of taxa on the co-abundance association signal

By construction, the MANOCCA association statistic is a weighted sum of the contribution from each pair of taxa considered on the overall co-abundance association statistics (see **Methods**). We derived those contribution weights for the age, sex, smoking and BMI signals, focusing on the genus level, which displayed the strongest association. As shown in Fig. S3a, most taxa display a non-zero contribution to the association, highlighting again the global effect of these factors on the microbiome composition. Nevertheless, these contributions show substantial heterogeneity, with a limited number of taxa pairs displaying outstanding weights as compared to the expected under the null (Fig. S3b). To assess potential links between effects on co-abundance and effects on relative abundance, we compared the MANOCCA weights against the univariate mean effect P-value associations derived using a standard univariate linear regression (see **Methods** and Supplementary Data 4). As shown in Fig. S3c, we observed a positive and significant correlation between the two results for all four variables (age, $P = 1.6 \times 10^{-12}$; sex, $P = 1.3 \times 10^{-40}$; smoking, $P = 2.1 \times 10^{-5}$; BMI, $P = 4.6 \times 10^{-15}$), suggesting a dual impact of these factors on the abundance and the co-abundance of many of these genera, in agreement with the existing literature[5–9,17,38–41]. Several top genera contributing to the co-abundance also passed a stringent Bonferroni correction threshold ($P < 8.7 \times 10^{-7}$) for univariate association. This includes *Bacteroides* ($P = 1.9 \times 10^{-7}$), *Coprococcus* B ($P = 7.1 \times 10^{-9}$), *Anaerotruncus* ($P = 8.5 \times 10^{-7}$), *Agathobacter* ($P = 3.0 \times 10^{-8}$), *Alistipes* ($P = 2.2 \times 10^{-8}$), and *Intestinimonas* ($P = 1.9 \times 10^{-9}$).

We next investigated the characteristics of the top 5% pairs of genera displaying the largest contribution to the variability in co-abundance at the family level. Out of 151 families, a subset of 10, 8, 11 and 7 overlapping sets of families, covered 50% or more of those top contributing genera with age, sex, smoking and BMI, respectively. Those key families include the ones with the highest relative abundance, *Lachnospiraceae* (23.6%), *Bacteroidaceae* (22.4%), *Ruminococcaceae* (7.8%), *Acutalibacteraceae* (6.4%), *Oscillospiraceae* (5.6%), but also some rare ones: *Eggerthellaceae* (0.5%), *Peptostreptococcaceae* (0.6%), *Muribaculaceae* (0.5%), and four unspecified Co-Abundance Groups (*CAG-74, CAG-508, CAG-272, CAG-138*) (Fig. 2a).

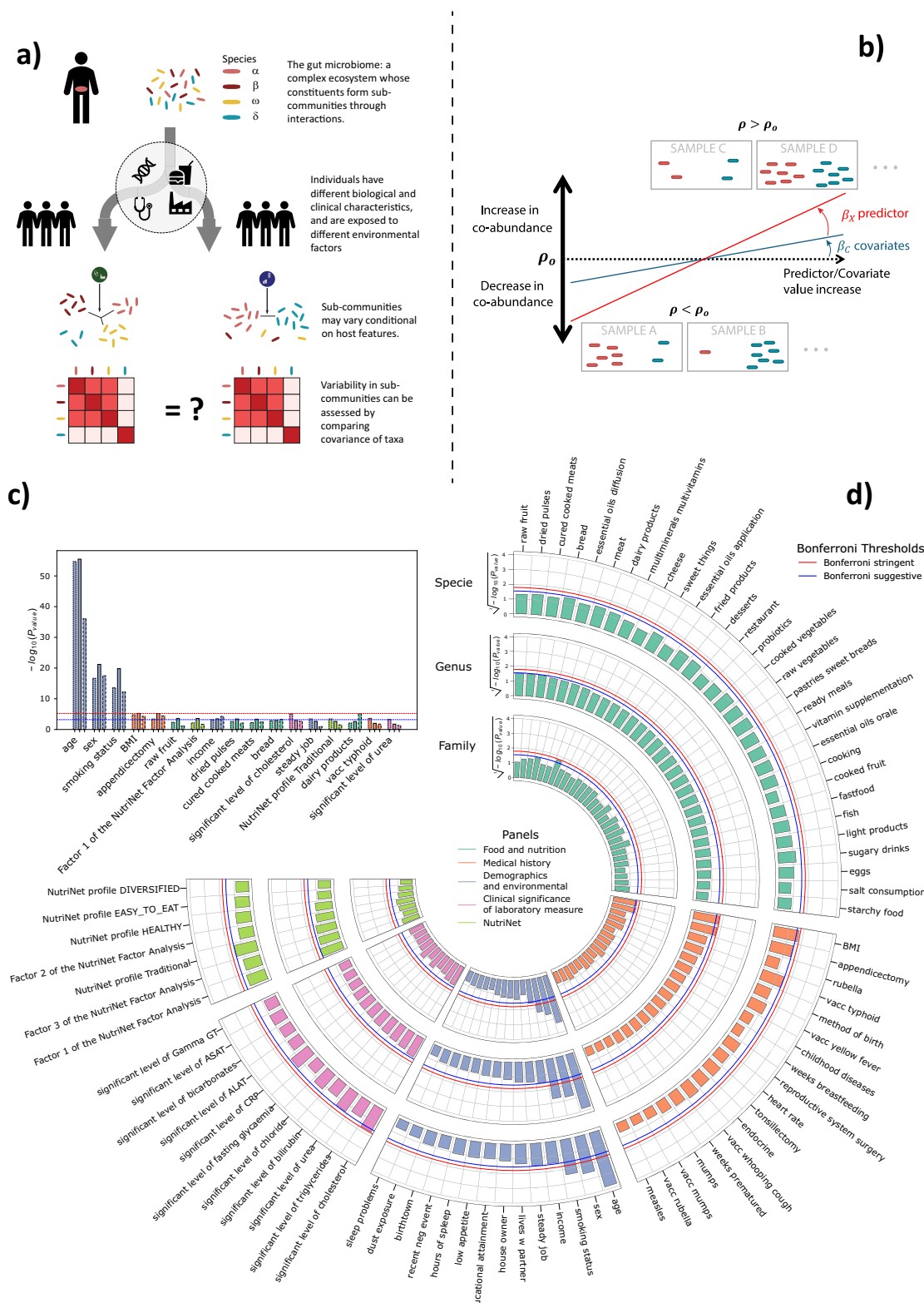

**Fig. 1 | Screening for environmental and clinical factors.** We performed an association screening between 80 environmental and clinical factors, derived from five panels of the Milieu Intérieur cohort, and the covariance of microbial taxa at the species, genus, and family levels. Panel (**a**) provides an overview of the analysis workflow, illustrating how environmental factors may influence microbiome co-abundances. Panel (**b**) presents a univariate example, demonstrating how changes in co-abundances are tested against variations in predictor and covariate values. Panel

(**c**) shows the $-\log_{10}P$ values for associations with at least one significant signal at any taxonomic level exceeding the suggestive Bonferroni threshold. Panel (**d**) displays the complete screening results on the cubic root scale for visualization purposes ($\sqrt[3]{-\log_{10}P}$). Results from MANOCCA are based on the optimal number of principal components. The blue dashed line represents the suggestive Bonferroni threshold, while the red dashed line indicates the stringent Bonferroni correction threshold accounting for all predictors and sets of PCs tested.

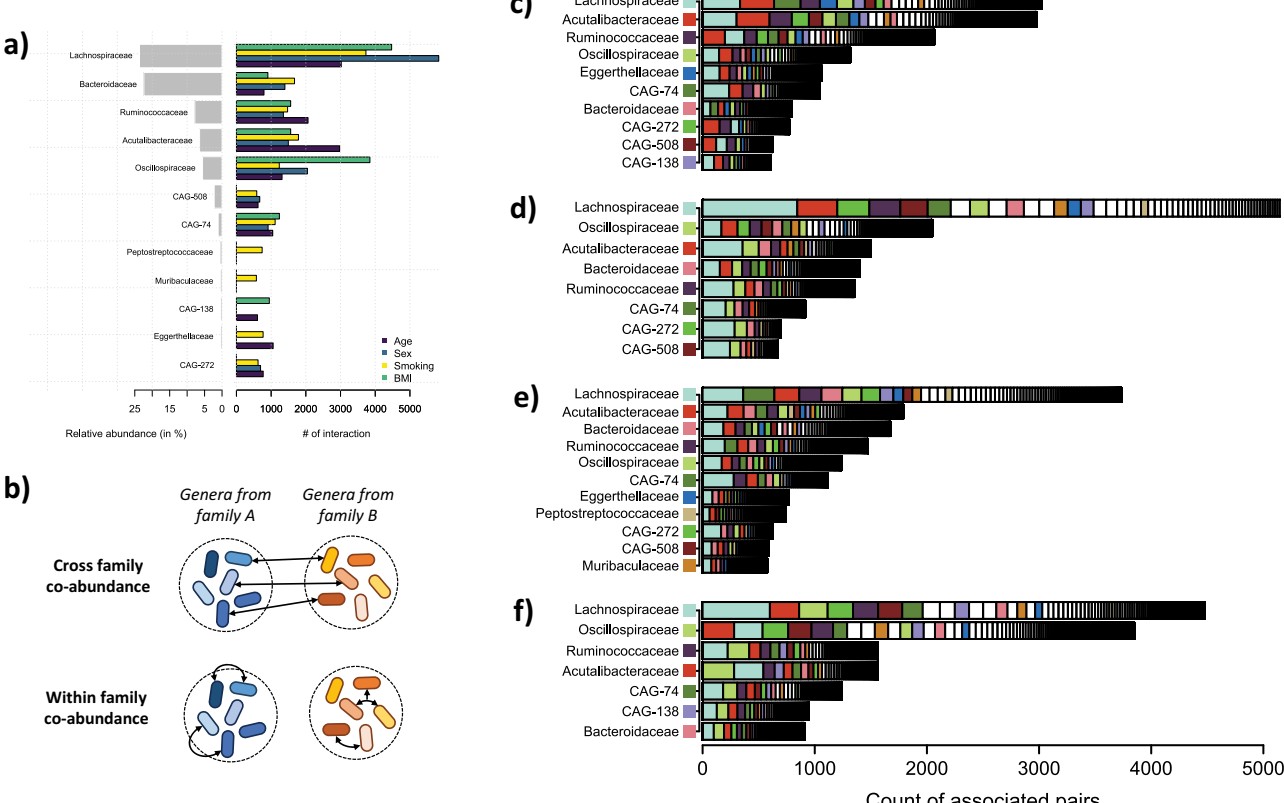

**Fig. 2 | Cross and inter-family interactions for age, sex, smoking and BMI.** We used the top 5% pairs of genera with the largest contribution to the covariance test to investigate whether variability in co-abundance involved genera from the same family or from different families. Panel (**a**) presents the relative abundance of the family from these top genera in the whole cohort (grey bar), and the number of interactions observed for genera from each of these families. Panel (**b**) illustrates the difference between cross-family and within family interactions. Detailed co-abundance pairs are presented for age (**c**), sex (**d**), smoking (**e**) and BMI (**f**). The Y axis represents the families of the top subset of genera involved, and that together explain up to 50% of the signal. The X axis shows the distribution of families for the associated genera, defined as a count per family. The length of the bar indicates how many pairs are involved for each top family. Colors were assigned only for the top families, and white blocs correspond to the unlisted categories.

While the representativity of families involved in co-abundance variability was somewhat proportional to their relative abundance, we noted several major differences. Some families, such as *Bacteroidaceae* are largely under-represented in the co-abundance signal. Conversely, co-abundance involving the *Oscillospiraceae* family are strongly impacted by all four factors, and by BMI in particular. Other families also display factor-specific enrichment, including *Peptostreptococcaceae* and *Muribaculaceae* with smoking, two families already reported to be associated with smoking status[42,43].

We examined the composition of the top contributing pairs of genera to determine whether they were involved in changes in interaction within the same family (intra-family co-abundance) or interaction between genera from different families (cross-family co-abundance) (Fig. 2b). Within-family co-abundance represented a small fraction of all interactions, with the vast majority of interactions taking place between genera of different families (Fig. 2c-f). Besides a few exceptions (e.g. variability in the co-abundance between taxa from *Oscillospiraceae* and *Acutalibacteraceae* families for BMI), we did not observe any marked pattern.

## Network of impacted taxa

We formed a network of co-abundance variation from the top 1000 pairs of genera contributing to the MANOCCA association signal with age, sex, smoking, and BMI (Fig. 3a, and **Supplementary Material**). Altogether, these 4000 pairs involved a total of 476 unique genera. As shown in Fig. 3b, there was a substantial overlap in pairs of co-abundant taxa impacted by sex and BMI ($N = 658$ pairs, approximately 66% of sex and BMI associated pairs), and age and smoking habits ($N = 306$ pairs, approximately 31% of age and smoking-associated pairs). Conversely, the overlap across the nine other pairs of factors was null or negligible. At the taxa level, a core of 200 genera were shared across all predictors (Fig. 3c). Other genera were evenly spread across factors, except for age and smoking which involved 49 (13%) and 54 (15%) genera specific to those two factors, respectively. Together, this suggests that the four factors partly control the interacting partners of this core genera. As shown in Fig. 3d, increased age and smoking are mostly associated with a decrease in co-abundances, with 86% and 75% of top pairs displaying negative associations with these two factors, respectively. BMI exhibited an opposing trend, with 72% of top pairs showing an increase in co-abundance with increasing BMI. The sex predictor displayed a more balanced distribution, with a 60% decrease and a 40% increase of co-abundance in males as compared to females.

Multiple patterns emerged when exploring the contributing genera. Those shared between smoking and age are especially enriched in the *Oscillospirales* order (e.g. *Massillioclostridium, CAG-180, CAG-1427, Marseille-P4683,* and *MGYG-HGUT-03297*), and consistently exhibited reduced co-abundances with the core taxa. Among genera unique to smoking, *Bacteroides A* genus was by far the most impacted, showing a reduction of co-abundances with many of the core taxa. Interestingly, the relative abundance of this common genus (detected in 99% of participants, Supplementary Data 2) was not associated with smoking status in our data (P-value from a linear regression equals 0.26, Supplementary Data 4). This suggests that smoking might only break some of its interactions with other genera without impacting the presence of this genus itself, highlighting the ability of our approach to detect taxa missed by standard abundance-based approaches. A subset of genera contributing to association with BMI, sex and smoking was enriched from the *Lachnospiraceae* family (*Ruminococcus*

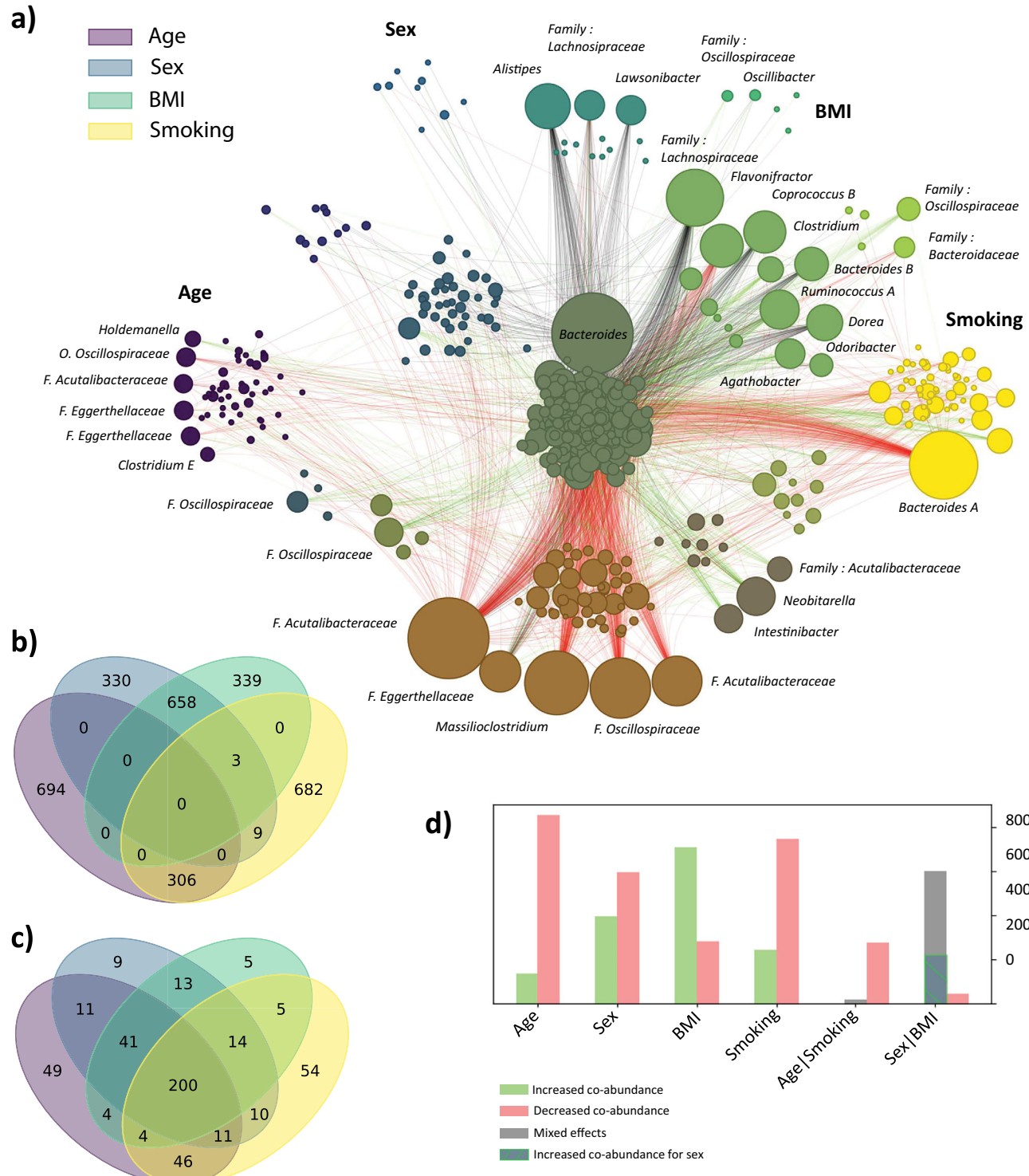

**Fig. 3 | Co-abundance network influenced by age, sex, smoking and BMI.** For the top four associated features from the MANOCCA (age, sex, BMI and smoking), we extracted the top 1000 contributing pairs of genera out of the 259,560 total products and derived the direction of effect of each predictor on the pair of co-abundance. We plotted the Venn diagram of shared pairs between each feature in (**a**) and the overlap in taxa in (**b**). In (**c**), we show the distribution of direction of effects per predictor, and for the age – smoking and sex – BMI intersections. We then used the pairs of features to derive a network of the changes in correlation with regard to each predictor. The node size, representing a genus, is proportional to its number of contributions with other genera, and edges link the top contributing pairs. The edge colors indicate the direction of effect with green indicating that an increase of the predictor drives an increase in co-abundance, red shows that an increase of the predictor drives a reduction in co-abundance and black indicates a mixed direction of effect for the overlapping predictors. The color of each node depends on how it is shared across the four predictors, and follows the structure of the (**b, c**) venn diagrams. Panel (**d**) displays the number of edges included in a single predictor (Age, Sex, BMI, Smoking) and by overlapping predictors (Age and Smoking, Sex and BMI), with in red the edges of reduced co-abundances and in green increased co-abundances. Grey edges indicate a mixed direction of effects for the overlapping predictors. Specifically for the overlap between Sex and BMI, the hashed area represents edges towards increased co-abundance for Sex and decreased co-abundance for BMI. Conversely, the grey part covers a decrease for Sex but an increase for BMI.

*A, Dorea, Coprococcus B, GCA-900066135, Agathobacter*), displaying both increased and decreased co-abundances across predictors. Both *Lachnospiraceae* relative abundance and co-abundance with other taxa have already been found to be associated with human diseases and obesity in particular[28,44].

To assess the relevance of the covariance-based co-abundances network impacted, we considered two alternative approaches. A naïve permutation-based approach, inspired from the existent[33], that produces an empirical comparison of pairwise covariance between all taxa (see **Methods**), and the commonly used SparCC[45] approach (see Supplementary Note 4 for a detailed description of the two approaches). Note that both the SparCC and the permutation-based approaches, like all existing method, are limited to binary predictors and use a threshold to define co-abundance in each group studied. Both methods are meant to detect significant differences in pairwise taxa correlations across values of a categorical predictor, and should in theory detect effect on co-abundance variability similar to those detected by MANOCCA. However, as showed in the simulation from Fig. S4a-d, permutation-based shows poor specificity as compared to MANOCCA, and SparCC shows the poorest performances in this simulation with almost no power. We applied both methods to the two binary predictors, sex and smoking, at the genus level and crossed the results with MANOCCA's top contributing products. The overlap between MANOCA and the two alternative methods was very modest, but highly significant with minimum *P*-values of $1 \times 10^{-145}$ and $2 \times 10^{-153}$ for sex and smoking respectively for the permutation approach (Fig. S4e and g), and $1 \times 10^{-70}$ and $1 \times 10^{-82}$ for sex and smoking respectively for the SparCC approach (Fig. S4f, h), thus, confirming that those three alternative methods do detect some similar network components.

## Prediction of individual features based on taxa correlation

Our framework is built out of a linear model where the covariance is defined at the individual level. This is a major advantage over existing correlation approaches[33], that allows for a range of complementary analyses. One particularly important extension is the possibility of training a predictive model of an outcome based on taxa covariance, so that the outcome in question can be predicted for any new individual based on its microbiome (and conversely). Here, we assessed the accuracy of MANOCCA to predict the four most associated features (age, sex, smoking and BMI), using taxa from the species, genus and family level and a 30-fold cross-validation. Accuracy was derived using squared-correlation ($r^2$) for continuous outcomes (age and BMI), and using the area under the receiver operating curve (AUC) for binary outcomes (smoking and sex). We compared the covariance-based prediction model against a standard linear model based on the relative abundance of each single taxa.

As showed in Fig. 4, the MANOCCA strongly outperforms the standard mean-based prediction model, being significantly more accurate in all scenarios we considered. Gain in prediction was especially large for age with up to a three-fold increase in power. The median of $r^2_{age}$ from the MANOCCA equals 0.27, 0.25 and 0.18 for models based on species, genus and family, respectively. In comparison, the mean-based model $r^2_{age}$ equal 0.10, 0.07 and 0.05, respectively. Prediction was also significantly higher for sex, with AUCs of 0.66, 0.64, and 0.64 for the mean-based model at the species, genus and family level, respectively, and AUCs of 0.67, 0.69 and 0.70 for the covariance-based model. This confirms the higher information content of co-abundance as compared to abundance, and demonstrates the validity of using covariance-based co-abundances for prediction purposes.

## Discussion

There is a strong rationale for studying changes in microbiome co-abundances. There is now increasing evidence that species form functionally coherent groups that work together to exploit the same resources from the local environment[30]. Studying those groups, rather than each single taxa, might help better understand the role of the microbiome in human health outcomes. With this same argument, it has already been proposed to study those groups through variability in the network of co-abundances[28,31].

Although simple in principle, the implementation of this objective can be challenging in practice. Here, we applied MANOCCA[34], a recently developed method, that enables formal statistical testing of associations between taxa covariance and any predictor - whether continuous, categorical or binary. We used it to investigate host features associated with gut microbiome co-abundance in 938 healthy individuals from the Milieu Intérieur cohort.

We identified highly significant associations between taxa co-abundance variability and age, sex, smoking status and body mass index (BMI). Except for BMI, these associations were detected at all three taxonomic levels studied: species, genus, and family. In comparison, mean-based multivariate and diversity-based analyses identified associations only with age, and one with sex at the family level, but at a much lower significance. For the four associated features, there was a significant correlation between contribution to co-abundance and univariate effect on relative abundance, suggesting that those features impact both the abundance and co-abundance of taxa. The network of top contributing genera shows that interaction variability was concentrated in a limited number of families. These interactions primarily occurred between genera of different families, rather than within the same family. The overlap of top contributing taxa over the four features was substantial, especially between age and smoking, and between sex and BMI, suggesting potentially shared mechanisms. Finally, we demonstrate that the MANOCCA framework can be used to build predictive models. In this study, we applied it to age, sex, smoking and BMI. For all features, the predictive power based on co-abundances was significantly and systematically higher than for a standard mean-based multivariate model, with up to a three-fold increase in r-squared for age.

Our study also has limitations. First, the approach is not applicable to microbiome data of small sample sizes. Despite the data reduction steps through principal component analysis, the number of PCs analysed should remain substantially smaller than the sample size, thus limiting the application to datasets of 100 participants or more. Hopefully, this will become less of an issue thanks to the increasingly large cohorts available. Second, the proposed approach does not model the compositional aspect of the data per se[46]. However, under reasonable assumptions, the change in covariance assessed by MANOCCA is independent of the marginal covariance. As a result, any potential bias due to compositional effects acts as an offset, without impacting our test. Additionally, when the dimension of the data is large enough, as for the analysis of species, genus or family conducted in this study, the compositional effect on correlation becomes negligible (Supplementary Note 3 and Fig. S4i-n). Third, new bacterial species databases have been published since our analysis, potentially offering novel insights. Fourth, we demonstrated that covariance can be used for prediction purposes. However, implementing such predictive models will require further exploration. As for prediction model based on relative abundance, some species might not be quantified in the targeted samples for prediction. This issue will likely be exacerbated when working with thousands of covariance terms. One possible solution is to develop sparse predictive models focusing on pairs of taxa that are fairly common, instead of using the entire covariance matrix. Furthermore, we used simple linear predictive models for both abundance and co-abundance. Future work might investigate the use of more complex methods[47] to combine the proposed covariance into prediction models.

Through the characterization of the links between variability in gut microbiome taxa co-abundance and healthy individual host factors, this study addresses three major limitations of the existent. First, the proposed approach allows for a formal statistical test of association between taxa co-abundance and both binary and continuous host features. In contrast, existing methods are restricted to ad hoc comparisons of inferred networks across a limited number of conditions. Second, the framework allows for covariate adjustment, so that the respective effects of correlated factors can be deciphered from one another. Third, the covariance-based approach provides a mean to derive a co-abundance metric at the individual level, allowing for a range of secondary analyses, including the development of co-abundance-based predictive models. Altogether, the proposed approach opens paths for various

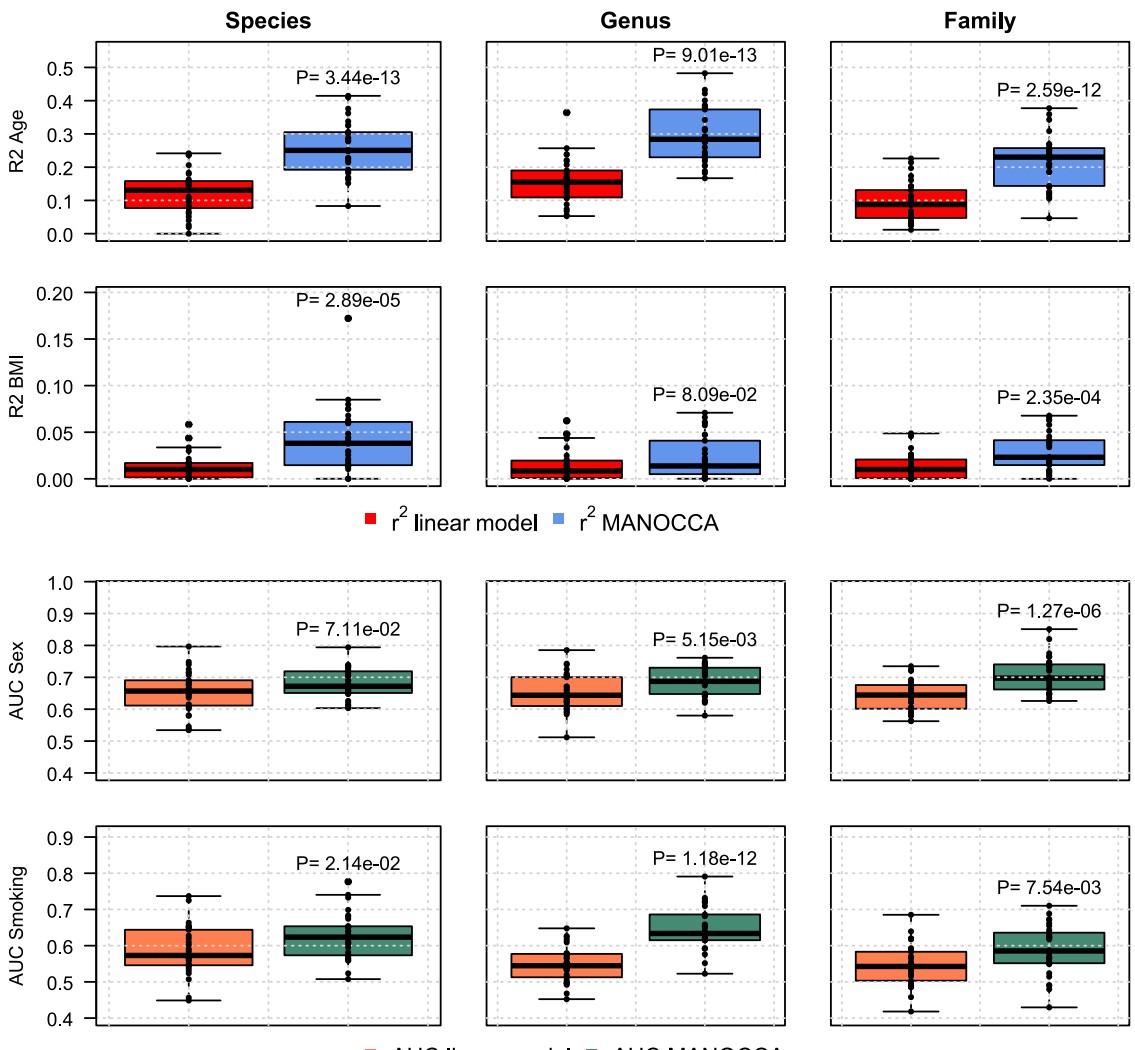

**Fig. 4 | Predictive power of covariance-based models.** We estimated the predictive power of co-abundance models (blue/green boxplots) as compared to the standard mean-based multivariate model (red/orange boxplots). For each of the four most associated features (age, sex, smoking and BMI) and the three lower taxonomic levels (species, genus, family), we derived the prediction accuracy using the squared-correlation ($r^2$) for continuous features and the area under the receiver operating curve (AUC) for binary feature. The analysis was done using a 30-fold cross validation, with the score being train in 90% of the data, and the $r^2$ being derived in the remaining 10%. The significance of the difference between the two models was tested using a two-sample two-sided *t*-test.

co-abundance analyses. It is highly complementary to recent efforts to develop experimental design to study co-abundance (e.g.[48]).

Overall, because the MANOCCA is built on a linear least squares framework, it inherits its assumptions and robustness, and can be extended to more complex scenarios using established solutions. Future works may examine the analysis of longitudinal data by modelling data structure through linear mixed model, or regularization to account for data specific issues. Conceptually, nothing prevents applying MANOCCA to other scenarios, such as taxa comparing taxa co-abundance between inflammatory bowel disease cases and controls. However, we highly recommend validating any new use case through calibration plots. Altogether, this approach can be used to produce new working hypothesis, and assess statistical evidence for effect on co-abundance from both observational and experimental data.

## Methods
### Milieu Interieur gut microbiome data
The Milieu Intérieur Consortium is a population-based cohort initiated in September, 2012[36]. It comprises 1,000 healthy volunteers, all recruited in the suburban Rennes area (Ille-et-Vilaine, Bretagne, France), with a 1:1 sex ratio

(500 males, 500 females) and an equal distribution across 5 decades (20 to <30 y, 30 to <40 y, 40 to <50 y, 50 to <60 y). The primary objectives of the MI Consortium are to define the naturally occurring variability in a healthy population's immune phenotypes and to characterize genetic, environmental, and clinical factors driving this variability. The cohort collected a broad range of variables, including genetic, genomic, and environmental data, on most participants. On their first visit the volunteers were also asked to fill in an extended form about socio-demographic, lifestyle and family health history, all recorded in an electronic case report form (eCRF). Gut microbiota composition was obtained from shotgun metagenomics sequencing, and taxonomic levels were reconstructed by summing the normalized abundances within a branch at a given level (Fig. S1), resulting in a total of 13,446 unique bacterial species. Further description of the data generation is provided in Supplementary Note 1.

### Covariance method
To test for changes in co-abundance we used the Multivariate Analysis Of Conditional Covariance Analysis test (MANOCCA) as previously published in Boetto et al.[34,49]. Briefly, the variability in the covariance between two standardized outcomes $Y_1$ and $Y_2$ can be investigated through the

element-wise product of those outcomes. For two outcomes $Y_1$ and $Y_2$ the covariance, or co-abundance, can be expressed as $\text{cov}(Y_1, Y_2) = \mathbb{E}[Y_1 Y_2] - \mathbb{E}[Y_1]\mathbb{E}[Y_2]$, and for standardized outcomes and a sample size $N$, it can be re-expressed as the average of the element-wise product across individuals: $\text{cov}(Y_1, Y_2) = \left(\sum_{i=1...N} Y_{1i} Y_{2i}\right)/N$. It follows that the effect of a predictor $X$ on $\text{cov}(Y_1, Y_2)$ can be tested using a standard least-squares regression framework where $X$ is treated as a predictor and the product $Y_1 Y_2$ as the outcome. One can easily demonstrate that, under reasonable assumptions, this test is independent of mean and variance effect[34]. Extending the method to more than two outcomes can be done through the following four steps: i) starting with $K$ centered outcomes $Y_1, \ldots, Y_K$, all the pairwise products are computed: $P_{ij} = Y_i Y_j$ for $i \in [\![1, K]\!]$ and $i < j$; ii) The $P_{ij}$ products are then mapped to the quantiles of a normal distribution using an inverse-rank normal transformation; iii) to reduce the dimension of product matrix, $P$ is then projected in a reduced latent space of dimension $p \ll \frac{k(k-1)}{2}$ using the Principal Components Analysis transformation: $PC_r = Q_r = \sum_{i=0}^{K} \sum_{j>i}^{K} \lambda_{ij}^{(r)} Y_i Y_j$, the resulting Principal Components (PC) are then mapped to the quantiles of a normal distribution using an inverse-rank transformation, and scaled. This gives, for $N$ considered individuals, a matrix $\mathbf{Q}$ of dimension $N \times p$ that we can use for the test. A detailed explanation of each step is available in Boetto et al.[34]. Finally, iv) given a scaled predictor $X$ and scaled covariates matrix $\mathbf{C}$, which can be continuous, categorical or binary, the test for association between the predictor and the covariance matrix can be conducted using a standard multivariate model: $\mathbf{Q} \sim X + \mathbf{C}$.

In this application, we varied the number of principal components used in MANOCCA from two to one hundred but limited the number of PC analysed for each predictor based on the guidelines provided in ref. 34, and used a stringent multiple testing significance threshold to account for the various number of PCs considered. Further description of the impact of dimension reduction on the MANOCCA test is provided in Supplementary Note 2.

## Contribution of taxa to covariance association signal
All the steps in the derivation of the statistical test are linear operations, which means that the contribution of features contributing to the MANOCCA association signal can be summed. Two types of contributions can be derived: the covariance contribution from each pair of taxa, and the sum of the covariance contribution assigned to each single taxon. The contribution of a given pair of taxa $i$ and $j$ to the covariance signal, $\phi(P_{ij})$, is defined as the square of the PCA loadings multiplied by the univariate association coefficient $\hat{\beta}^2$ of the corresponding principal components with the considered predictor: $\phi(P_{ij}) = \sum_{r=1}^{p} \hat{\beta}_r^2 \left(\lambda_{ij}^{(r)}\right)^2$, where $\lambda_{ij}^{(r)}$ is the loading of the $ij$ pair of taxa for PC $r$, and $p$ is the total number of PCs included in the analysis. The single taxa contribution, $\psi(Y_i)$, can be derived by summing its contributions across all pairs: $\psi(Y_i) = \sum_{j=1}^{K-1} \phi(P_{ij})$, with $j \neq i$, and $K$ is the total number of pairs.

## Environmental association screening using MANOCCA
We applied MANOCCA to identify environmental factors associated with a change in covariances between taxa at the family, genus and species level. Milieu Interieur volunteers were asked to fill in a questionnaire of 44 pages, covering multiple panels such as demographics, lifestyle, and vaccination history. We selected the most relevant panels for the study, leading to the selection of 102 environmental factors. Among them was included diet information collected as part of the Nutrinet study[50]: the top three factors from the Nutrinet factors analysis, and the Nutrinet profiles were binarized to yes/no. We filtered out variables with more than half of the sample size in missing values, or a binary predictor with frequencies smaller than 5%. For categorical predictors displaying highly skewed distributions, outliers, defined as value three more standard deviation away from the mean were merged with a lower occurring category. A total of 80 environmental factors remained for analysis. After filtering, we ended up with a cohort of 938

individuals with complete shotgun sequencing, age, sex and body mass index (BMI) data. For the genus and family levels, we kept taxa abundant in at least 5% of the cohort, leading to a drop from 1192 genera to 718 genera, and 216 families to 151 families. At the specie level, to avoid having too many species with regard to the sample size, we set the threshold to 40% of the cohort leading to a drop from 3885 species to 675 species.

## Significance thresholds and sensitivity analyses
We used two significance level to account for multiple testing. A very stringent Bonferroni corrected threshold, $B_{stringent} = 0.05/(80 \times 100) = 6.25 \times 10^{-6}$, that cumulate the total number of test (80 host feature and 100 principal component (PC) models) without accounting for the strong correlation between the PC models, and a suggestive threshold, $B_{suggestive} = 0.05/80 = 6.25 \times 10^{-4}$, that assume strong correlation across the PCs models.

## Comparison with MANOVA and alpha diversity
For comparison purposes, we considered three alternative multivariate methods: a standard MANOVA and the alpha diversity, using the Shannon and Simpson indexes. The screening methodology was the same as the one used for MANOCCA, though some pre-processing adjustments were made to match the expected assumptions of each method. The MANOVA was applied to the taxa relative abundances from a given phylogenic level, which was processed following standards from the literature[51]: proportion followed by arcsin root transformation followed by a scaling. With $\mathbf{Y}$ the matrix of resulting taxa, $X$ the considered predictor, and $\mathbf{C}$ a matrix of covariates. We applied the Wilk's lambda test : $\mathbf{Y} \sim X + \mathbf{C}$, and in more details with $\widetilde{\mathbf{Y}} = \mathbf{Y} - \beta_c C$ the residual matrix after adjustment from the covariates, $\hat{\beta} = (X^T X)^{-1}(X^T \widetilde{Y})$ the regression coefficient and $N$ the sample size, we could compute a $P$-value for the statistic : $\det\left(\widetilde{Y}^T \widetilde{Y} - N\hat{\beta}\hat{\beta}^T\right)/\det(\widetilde{Y}^T \widetilde{Y}) \sim F(p, N - p - 1)$.

For the alpha diversity indexes, the raw abundances were used to the corresponding metric : $\alpha_{Shannon} = \sum_{i=1}^{N} \frac{x_i}{\sum_{j=1}^{N} x_j} \log\left(\frac{x_i}{\sum_{j=1}^{N} x_j}\right)$ and $\alpha_{Simpson} = 1 - \sum_{i=1}^{N} \frac{x_i(x_i-1)}{\left(\sum_{j=1}^{N} x_j\right)\left(\left(\sum_{j=1}^{N} x_j\right)-1\right)}$. The resulting $\alpha$ was tested in a standard univariate linear regression adjusted for the covariates: $\alpha \sim \delta_X X + \delta_C C$. The effect of $X$ was assessed using a Wald test to the $\hat{\delta}_X$.

## Deriving the covariance network
Networks of variation in the covariance were built using the top 1,000 co-abunding pairs derived using the MANOCCA test. In representing the network, we included three parameters: the total number of connections (qualified through the node size), the actual pairwise taxa connection (edges in the graph), and the direction of host factor effect on the covariance (decrease or increase of co-abundance). For each pair $Y_i Y_j$ adjusted for covariates $C : \widetilde{Y_i Y_j} = Y_i Y_j - (C^T C)^{-1} C^T Y_i Y_j C$, the direction of effect was derived using the sign of the regression coefficient for the predictor $X$ : $\beta_{ij} = (X^T X)^{-1} X \widetilde{Y_i Y_j}$. For shared pairs with mixed direction of effect, the edge was colored in black. To facilitate the reading of the network, we coloured the node conditional on the association with each of the four predictors of interest, or shared among them. The 'viridis' cmap from *matplotlib* was used as colour scheme, with each shared taxa being a combination of original predictor colours.

## Comparison with other network-based approaches
For comparison purposes, we derived a permutation-based network inference approach for binary predictor, which we used to validate the MANOCCA network. In brief, we derive the pairwise covariance matrix for each categorical value, and then derive the empirical distribution of the correlation under the null by simulating $N_{permutations}$ covariances after shuffling the abundances of a bacterial taxa for each individual. Using a fixed detection threshold, we can then select pairs of taxa with extreme

covariances. Since we are interested in variability of the covariance, we only keep the pairs uniquely detected across all values of the given predictor. When applied to compare results from the sex and smoking analyses, we ran 100,000 permutations, and retrieved the unique pairs detected in either group (women vs men, and non-smoker vs ever smoke). We also ran the SparCC[45] correlation analysis, as this approach is commonly used and performed relatively well in a review of existing approaches[33]. We ran SparCC using the recommended parameters, deriving the P-values using 1000 permutations, on both the simulated data (Fig. S4c) and the real data (Fig. S4, h). Further descriptions of both approaches are provided in the Supplementary Note 4.

### Using co-abundance for prediction purposes

We assessed the performances of a predictive model based on covariance across taxa. The implementation of a predictive model follows the standard used for multivariate linear model. For a given outcome $A$ to be predicted, the estimated coefficients between $A$ and $PC_{i=1...L}^{(train)}$ obtained in a training dataset from MANOCCA, $\hat{\boldsymbol{\beta}} = (\hat{\beta}_1 \ldots \hat{\beta}_L)$ are projected on the principal component from an independent test dataset and summed up to to form a predictive score $S = \sum_{i=1}^{L} \hat{\beta}_i PC_i^{(test)}$. Note that the dimensionality of the covariance data and the principal component analysis (PCA) step make the implementation slightly more complex. In particular, the principal components derived on the same variables for two independent samples might not always match, with structure in the data being capture by different components. To avoid this issue, PCA is not applied in the test data. Instead, $PC_i^{(test)}$ are derived by projecting out the loadings from the training sets: $PC_i^{(test)} = \sum_1^L \lambda_j^{(i)} X_j$, where $\lambda_j^{(i)}$ is the loading of variable $PC_i^{(train)}$ for product of taxa $i$ obtained in the train data. It also implies that the test dataset should have the same dimension (i.e. approximately the same list of taxa) as the train dataset.

We applied this approach for the prediction of age, BMI, smoking and sex using taxa from the three lowest taxa levels (species, genus and family), using a 30-fold cross-validation, and without including other factors as covariate. For each of the 30 cross-validation, the dataset was randomly split into two independent sets: a training set including 90% of the data and a test set including the 10% remaining samples. We measured the accuracy of the predictive model using squared-correlation for continuous outcomes, derived as $cor\left(S, A^{(test)}\right)^2$, and using the area under the receiver operating curve (AUC) for binary outcomes. The AUC is a common metric to quantify the predictive power of binary outcome. It equals the probability of correctly classifying a random sample from the test data.

### Statistics and reproducibility

All statistical analyses were performed using the MANOCCA framework for multivariate covariance analysis, the MANOVA for multivariate additive analysis, and otherwise a standard linear regression. All models were fitted using linear regression under least squares assumptions.

### Reporting summary

Further information on research design is available in the Nature Portfolio Reporting Summary linked to this article.

### Data availability

The dataset supporting the conclusions of this article is available in the European Genome-Phenome Archive under accession code EGAC00001001785. Figure 1 was generated using data from Supplementary Data 1 and Supplementary Data 3. Figure 2 was generated using data from Supplementary Data 2 and Supplementary Data 3. Figure 3 was generated using data from Supplementary Data 2 and Supplementary Data 3. Figure 4 was generated using data from Supplementary Data 4.

### Code availability

All code is available in Python and R at: https://gitlab.pasteur.fr/statistical-genetics/manocca and at https://doi.org/10.5281/zenodo.16945401.

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

## Acknowledgements
This research was supported by the Agence Nationale pour la Recherche (ANR-20-CE15-0012-01). This work has been conducted as part of the INCEPTION program (Investissement d'Avenir grant ANR-16-CONV-0005). The Milieu Interieur consortium was also supported by the Agence Nationale pour la Recherche (ANR-10-LABX-69-01). More specifically, authors C.B., V.B.R., L.H., A.F., A.A., M.B., S.P.K., L.Q.M., H.S. and H.A. were funded using ANR-20-CE15-0012-01. Authors E.P., D.D. and L.Q.M. were funded using ANR-10-LABX-69-01. The funders had no role in study design, data collection and analysis, decision to publish, or preparation of the manuscript.

## Author contributions
C.B. performed analyses and drafted the manuscript. V.B.R. worked on data selection and preprocessing, L.H. worked on bioinformatics pipelines, A.F. and A.A. contributed addressing methodological questions, E.P., M.B., S.P.K. worked on the microbiome data processing and analysis, D.D., L.Q.M. and H.S. supervised the microbiome data generation and analyses, H.A. ran some analyses, supervised the study and drafted the manuscript.

## Competing interests
The authors declare no competing interests.

## Additional information

**Milieu Intérieur Consortium**

**Laurent Abel**[9], **Andres Alcover**[10], **Hugues Aschard**[10], **Philippe Bousso**[10], **Nollaig Bourke**[10], **Petter Brodin**[11], **Pierre Bruhns**[12], **Nadine Cerf-Bensussan**[12], **Ana Cumano**[13], **Christophe D'Enfert**[13], **Ludovic Deriano**[13], **Marie-Agnès Dillies**[13], **James Di Santo**[13], **Gérard Eberl**[13], **Jost Enninga**[13], **Jacques Fellay**[13], **Ivo Gomperts-Boneca**[11], **Milena Hasan**[11], **Gunilla Karlsson Hedestam**[11], **Serge Hercberg**[14], **Molly A. Ingersoll**[15], **Olivier Lantz**[16], **Rose Anne Kenny**[17], **Mickaël Ménager**[12], **Frédérique Michel**[17], **Hugo Mouquet**[17], **Cliona O'Farrelly**[17], **Etienne Patin**[12], **Sandra Pellegrini**[12], **Antonio Rausell**[12], **Frédéric Rieux-Laucat**[12], **Lars Rogge**[18], **Magnus Fontes**[18], **Anavaj Sakuntabhai**[19], **Olivier Schwartz**[19], **Benno Schwikowski**[19], **Spencer Shorte**[19], **Frédéric Tangy**[19], **Antoine Toubert**[19], **Mathilde Touvier**[14], **Marie-Noëlle Ungeheuer**[20], **Christophe Zimmer**[20], **Matthew L. Albert**[20], **Darragh Duffy** [iD][4] **& Lluis Quintana-Murci**[2,5]

[9]Hôpital Necker, Paris, France. [10]Trinity College, Dublin, Ireland. [11]Karolinska Institutet, Stockholm, Sweden. [12]INSERM UMR 1163 – Institut Imagine, Paris, France. [13]EPFL, Lausanne, Switzerland. [14]Université Paris 13, Paris, France. [15]Institut Cochin and Institut Pasteur, Paris, France. [16]Institut Curie, Paris, France. [17]Trinity College Dublin, Dublin, Ireland. [18]Institut Roche, Paris, France. [19]Hôpital Saint-Louis, Paris, France. [20]In Sitro, San Francisco, California, US.

