## [Transparent Peer review file · Communications Biology]

A landscape of the human gut microbiome communities in healthy individuals

Corresponding Author: Dr Christophe Boetto

Version 0:

Reviewer comments:

Reviewer #1

(Remarks to the Author)

The study leverages the MANOCCA framework to explore how host factors influence the covariance structure (co-abundance network) of the human gut microbiome in a large cohort of 938 healthy French adults. Unlike conventional abundance-based or diversity metrics, MANOCCA tests for changes in relationships (covariances) between microbial taxa, identifying patterns of co-abundance that are often missed by existing methods—thereby capturing more complex, ecologically meaningful microbial interactions. This statistical framework notably addresses the limitations of existing co-occurrence approaches that are restricted to categorical predictors, making it a valuable and much-needed methodological contribution. I have a couple of major, and minor comments I encourage the authors to address:

Major comments:

1. The authors filtered taxa present in >5% of participants at genus/family levels, and >40% at species level. Is MANOCCA sensitive to taxonomic resolution and data filtering thresholds?
2. While MANOCCA excels in healthy adults, microbiome communities in disease states often have sparser, less stable co-abundances. Would MANOCCA maintain its statistical power and calibration under such disrupted microbial ecosystems?
3. Current implementation assumes unstructured, cross-sectional data. However, many microbiome studies now collect repeated measures over time (longitudinal designs), across tissues, or within families/nested designs. Can MANOCCA be extended to handle longitudinal or hierarchical data structures, where repeated measurements or clustering are present?

Minor comments:

Line 349: The sub-numbering currently reads iv) — this should be corrected to iii).

Reviewer #2

(Remarks to the Author)

Boetto et al. demonstrate a novel analysis of microbial community composition data on a large dataset of human gut microbiome samples. The analysis technique is interesting and valuable, and its demonstration on a large dataset is effective in showcasing its power. Overall, I think that the manuscript is of very high quality. Pairwise co-abundance based techniques are very popular in microbiome research because of their easy interpretability but lack solid metrics for reliability. Thus, a focus on the predictive power of microbial co-abundance is welcome. Furthermore, Boetto et al.'s technique has a distinct focus on the differences in co-abundance across environmental conditions, which is often of interest to researchers. However, I have a few questions about the presentation of the technique and its comparison to existing techniques.

1. As I understand it, the technique presented by the manuscript (MANOCCA) has a fundamental difference with previous co-abundance analysis. Previous techniques, including SparCC, either explicitly (as is the case for the Gaussian LASSO method of Speic-Easi, for example) or implicitly assume that taxa composition in a sample is a random vector drawn from some single underlying distribution and the covariance matrix of this distribution is what is being inferred. Meanwhile,

MANOCCA seems to assume that the underlying distribution varies across some environmental parameter (a good assumption) and tries to infer this relationship. This needs to be better clarified in the introduction, because without that key difference highlighted the results are very hard to interpret.

2. I think that the above distinction also means that MANOCCA has a subtly different goal than previous techniques, because it will fail to identify strongly interacting taxa whose interactions is constant across environmental conditions.

3. The use of existing techniques as predictors in a linear model makes sense as a direct comparison to MANOCCA, but I would be interested to know if it would be more effective to use the likelihood of test samples under the distributions inferred by training samples (one could assume, for example, log-normal distributions with the estimated covariance matrices).

Reviewer #3

(Remarks to the Author)

The authors use their previously developed MANOCCA approach, a multivariate method to analyze taxa co-abundance using data from the Milieu Intérieur study cohort. They find that the factors of age, sex, smoking status, and BMI have the largest effect on the covariance of taxa. From there they show the size of the association for each factor, both within and across families. The predictive power of the MANOCCA approach outperformed the standard multivariate linear model. The presented data add to the complexity of the microbiome to other health-related factors and should be interesting to the field.

1. Both the previous publication (MANOCCA: a robust and computationally efficient test of covariance in high-dimension multivariate omics data) and the current manuscript provide the same code in the same gitlab repository. Since the acceptance of the previous publication, it seems that minimal additional changes have been made to this codebase. Please describe specific changes/improvements that have been made that are the basis for this new manuscript. If there is new code specific to the current work, please make it available.

2. There are problems with the submitted code: 'Example_manocca.ipynb' does not run. The keyword argument 'plot' was removed in commit 96489f9e, but it is still being used as an argument in multiple locations within the jupyter notebook. Likewise, manocca.r returns an error, because the files in the following lines are not provided:

```
df_shot <- read.csv('../notebooks/MI_screening/for_R/shot_data.csv')
df_covariates <- read.csv('../notebooks/MI_screening/for_R/covariates_age_sex_bmi.csv')
df_tabac <- read.csv('../notebooks/MI_screening/for_R/tabac.csv')
```

3. Sections with identical wording appear in their recent publication and this submitted manuscript. Appropriate referencing of their previous work needs to be done and methods sections (for example lines 298-305) need to be adjusted.

4. Grammatical mistakes should be corrected throughout the entire manuscript; for example, in the abstract it should be:

Line 19: functional communities whose effect
Line 20: underlying variability in these communities
Line 21: links between the gut microbiome
Line 25: and conversely, an increased body mass index

5. When possible, render figures in a vector graphics format. This is most important for those figures with small text.

6. Lines 133-134: '9, 7, 9 and 6' families are mentioned, but in Figure 2c-f there are 10, 8, 11, and 7 families depicted.

7. Line 579: mislabeling of age(c), sex(d), smoking(e) and BMI(f)

8. The squared correlation has a lowercase 'r' on lines 211, 216, and 218, but an uppercase 'R' on lines 606 and 608. The exponent is not superscripted for 'R²' on line 608.

9. Use the same coloring scheme for age, sex, smoking, and BMI in Figure 2a and Figure 3a-c.

10. Supplementary Material Line 37: A Kraken-GTDB database from 2019 is used. It should be considered to utilize a newer database that could lead to some improvement.

11. Figure 4: Reword AUC Tabac to AUC Smoking Status

Version 1:

Reviewer comments:

Reviewer #1

(Remarks to the Author)

I thank the authors for addressing my comments. In the revised manuscript, all of my comments has been addressed and I have no further comments.

Reviewer #2

(Remarks to the Author)

My two major points were addressed well.

For my third point, I was referring to the use of "other network based approaches". These approaches can be made into classifiers simply by comparing the probability density function at a data point from the inferred distributions for each class. This might be of interest, but in my opinion its inclusion is not necessary for publication of the manuscript.

Reviewer #3

(Remarks to the Author)

The authors have addressed all of my issues. I have no further comments.

Reviewer 1

1. The authors filtered taxa present in >5% of participants at genus/family levels, and >40% at species level. Is MANOCCA sensitive to taxonomic resolution and data filtering thresholds?

The motivation for filtering taxa was twofold. First, as guessed by the reviewer, it aims at avoiding model instability in the presence of sparse data. Removing such variables is common practice when applying univariate and multivariate linear regression. Second, in the present analysis, we were mainly interested in looking at co-abundance between common species variations in healthy individuals rather than rare species.

In order to address the reviewer question and provide the reader a better understanding of the approach's behavior conditional on the data filtering, we conducted multiple additional simulations using sparse data. The results from these simulations have been added to Figure S1 (and copied below). They display QQ-plots when increasing data sparsity from 100% to 5% for a model testing a random predictor against a squared multivariate normal mimicking microbiome data distribution. In those simulations, MANOCCA maintained its calibration for all amount of sparsity considered. Note that concurrently, Figure S2e-h points towards a convergence of p-values when increasing in taxa included in the analysis, suggesting that maximizing the inclusion of taxa may increase power. We clarified that aspect in the main text.

2. While MANOCCA excels in healthy adults, microbiome communities in disease states often have sparser, less stable co-abundances. Would MANOCCA maintain its statistical power and calibration under such disrupted microbial ecosystems?

This is an excellent point. Conceptually, nothing prevents from applying MANOCCA in other contexts, even outside of microbiome studies, and applying the approach in disease cases is one of our upcoming objective. The simulations provided in response to comment #1 (Figure S1c-d) partly address the question of the approach robustness in the presence of sparse data. Nevertheless, given the novelty of the approach, we now clarified in the discussion that conducting calibration simulations is required:

“Overall, because the MANOCCA is built on a linear least square framework, it inherits its assumptions and robustness, and can be extended to more complex scenarios using established solutions. Future works may examine the analysis of longitudinal data by modelling data structure through linear mixed model, or regularization to account for data specific issues. Conceptually, nothing prevents from applying MANOCCA to other scenarios such as taxa co-abundance comparison between inflammatory bowel disease case and controls, though we highly recommend any new use case to be validated through calibration plots. Altogether, this approach can be used to produce new working hypothesis, and assess statistical evidence for effect on co-abundance from both observational and experimental data.”

3. Current implementation assumes unstructured, cross-sectional data. However, many microbiome studies now collect repeated measures over time (longitudinal designs), across tissues, or within families/nested designs. Can MANOCCA be extended to handle longitudinal or hierarchical data structures, where repeated measurements or clustering are present?

Thanks to its linear framework the MANOCCA test can theoretically be extended to longitudinal data using linear mixed model (LMM), which are commonly used to model structure in the data. However, the applicability of such a solution will have to examine carefully given the sensitivity of LMM and the required sample size to ensure a valid estimation of the variance components. We added some comments on that aspect in the discussion:

“Overall, because the MANOCCA is built on a linear least square framework, it inherits its assumptions and robustness, and can be extended to more complex scenarios using established solutions. Future works may examine the analysis of longitudinal data by modelling data structure through linear mixed model, or regularization to account for data specific issues. Conceptually, nothing prevents from applying MANOCCA to other scenarios such as taxa co-abundance comparison between inflammatory bowel disease case and controls, though we highly recommend any new use case to be validated through calibration plots. Altogether, this approach can be used to produce new working hypothesis, and assess statistical evidence for effect on co-abundance from both observational and experimental data.”

Minor comments:

Line 349: The sub-numbering currently reads iv) — this should be corrected to iii).

Thank you for pointing that out, the paragraph was restructured to better depict the steps.

Reviewer 2

1. As I understand it, the technique presented by the manuscript (MANOCCA) has a fundamental difference with previous co-abundance analysis. Previous techniques, including SparCC, either explicitly (as is the case for the Gaussian LASSO method of Speic-Easi, for example) or implicitly assume that taxa composition in a sample is a random vector drawn from some single underlying distribution and the covariance matrix of this distribution is what is being inferred. Meanwhile, MANOCCA seems to assume that the underlying distribution varies across some environmental parameter (a good assumption) and tries to infer this relationship. This needs to be better clarified in the introduction, because without that key difference highlighted the results are very hard to interpret.

This is indeed a very important point and we thank the reviewer for highlighting this gap in our manuscript. As mentioned by the reviewer, the MANOCCA aims at identifying factors associated with a change in the joint distribution of a multivariate outcome, whereas most other methods separate data base on a binary or categorical factors of interest and estimate the levels of correlations within each group. We have now added a whole introduction paragraph, which we hope better introduce the motivations and difference of the MANOCCA test against the existent. The paragraph reads as follows:

“The recently developed MANOCCA⁴³ test is a formal statistical framework to test the effect of both categorical and continuous predictors on the covariance matrix of a multivariate outcome –a metric directly proportional to the co-abundance– addressing all these methodological gaps. Briefly, the MANOCCA models pairwise taxa covariances at the individual level (i.e. each individual has its own taxa covariance matrix) and use them as dependent variables in a standard multivariate regression to evaluate their association with predictors of interest. Existing tools⁴⁴ assume the observed covariances for a given discrete condition (e.g. male or female) are drawn from a single underlying distribution and aim at inferring that distribution. In comparison, the MANOCCA treats the covariances as random variables that varies across environmental factors, and aims at estimating the effect of those factors. Note that by construction, the test is independent of the baseline covariance, and does not aim at identifying strongly co-abundant taxa whose interactions is constant across environmental conditions.”

We also added two panels to Figure 1, to illustrate the principle of the proposed test (panel a)) and the univariate covariances tested (panel b)).

We hope those additions clarifies the motivations of our analysis.

2. I think that the above distinction also means that MANOCCA has a subtly different goal than previous techniques, because it will fail to identify strongly interacting taxa whose interactions is constant across environmental conditions.

This is also a very valid point. The MANOCCA does not aim at identifying interactions that are constant across environmental conditions and only assess variations in covariance regardless of the baseline correlation. Still, our application assumes that stronger correlations are expected to be more subject to variations. Additionally, as pointed in the first method paper describing MANOCCA, there can be orthogonality between additive and covariance signal making the MANOCCA detect a different kind of signals in comparison to existing additive tests.

This point was clarified in the introductory paragraph and the Fig. 1 panels which were added as part of the response to comment #1.

[figure redacted]

3. The use of existing techniques as predictors in a linear model makes sense as a direct comparison to MANOCCA, but I would be interested to know if it would be more effective to use the likelihood of test samples under the distributions inferred by training samples (one could assume, for example, log-normal distributions with the estimated covariance matrices).

We assume the reviewer is referring to the comparison of the MANOCCA with the standard MANOVA. We understand that the reviewer suggests that the conditional null distribution of the covariance matrix could be derived from training samples, and further be used to build a test by comparing the distribution observed conditional on a predictor value against this null. Although we envisioned how such an approach can be used to derive a test for categorical predictors, it is unclear how one can apply it to assess the effect of a continuous predictor, as deriving the conditional null distribution would be difficult to specify. We would happily investigate this point further but additional guidance would be welcome.

Reviewer 3

1. Both the previous publication (MANOCCA: a robust and computationally efficient test of covariance in high-dimension multivariate omics data) and the current manuscript provide the same code in the same gitlab repository. Since the acceptance of the previous publication, it seems that minimal additional changes have been made to this codebase. Please describe specific changes/improvements that have been made that are the basis for this new manuscript. If there is new code specific to the current work, please make it available.

We thank the reviewer for catching this gap. We have updated the gitlab to ease the configuration and usage of our tool. In particular, the current python repository now has a conda install setup (recommended), a pip requirements and a docker repository setup. All three options can be used to run the pipeline. Additionally the 'Example_manocca.ipynb' was improved and the deprecated code in the R directory was removed.

2. There are problems with the submitted code: 'Example_manocca.ipynb' does not run. The keyword argument 'plot' was removed in commit 96489f9e, but it is still being used as an argument in multiple locations within the jupyter notebook.

Likewise, manocca.r returns an error, because the files in the following lines are not provided:

```
df_shot <- read.csv('../..//notebooks/MI_screening/for_R/shot_data.csv')
```

```
df_covariates <- read.csv('../..//notebooks/MI_screening/for_R/covariates_age_sex_bmi.csv')
```

```
df_tabac <- read.csv('../..//notebooks/MI_screening/for_R/tabac.csv')
```

Again, we thank the reviewer for the thorough review of our work. In response to this comment and the previous one, all code was updated to be easier to use and run. All issues mentioned, both in R and python, have now been fixed.

3. Sections with identical wording appear in their recent publication and this submitted manuscript. Appropriate referencing of their previous work needs to be done and methods sections (for example lines 298-305) need to be adjusted.

We added an appropriate referencing of our previous work, and updated the method section.

4. Grammatical mistakes should be corrected throughout the entire manuscript; for example, in the abstract it should be:

Line 19: functional communities whose effect

Line 20: underlying variability in these communities

Line 21: links between the gut microbiome

Line 25: and conversely, an increased body mass index

For this comment and the following ones, we thank reviewer 3 for his very thorough proof reading of our manuscript. We made all the necessary modifications following those comments, and thoroughly checked for additional mistakes.

5. When possible, render figures in a vector graphics format. This is most important for those figures with small text.

We thank the author for this useful suggestion. All main figures have been converted to an SVG format.

6. Lines 133-134: '9, 7, 9 and 6' families are mentioned, but in Figure 2c-f there are 10, 8, 11, and 7 families depicted.

The correct number of families was updated in the text.

7. Line 579: mislabeling of age(c), sex(d), smoking(e) and BMI(f)

We corrected the mislabeling in the legend.

8. The squared correlation has a lowercase 'r' on lines 211, 216, and 218, but an uppercase 'R' on lines 606 and 608. The exponent is not superscripted for 'R2' on line 608.

We standardized our notation to a lower 'r'.

9. Use the same coloring scheme for age, sex, smoking, and BMI in Figure 2a and Figure 3a-c.

We matched the coloring scheme for both figures.

10. Supplementary Material Line 37: A Kraken-GTDB database from 2019 is used. It should be considered to utilize a newer database that could lead to some improvement.

This is a valid point and new tools are regularly getting published. Using a new database here would require re-running almost all of the analyses of the current paper. We had to decide on a specific database freeze when initiating the analysis. Nevertheless, we now acknowledge this limitation in the discussion, and we will use newer databases in future analyses.

“Third, new bacterial species databases have been published since our analysis, potentially offering novel insights.”

11. Figure 4: Reword AUC Tabac to AUC Smoking Status

We corrected that typo.